# Hearing as adaptive cascaded envelope interpolation

Etienne Thoret [1,2✉], Sølvi Ystad [3] & Richard Kronland-Martinet [3]

The human auditory system is designed to capture and encode sounds from our surroundings and conspecifics. However, the precise mechanisms by which it adaptively extracts the most important spectro-temporal information from sounds are still not fully understood. Previous auditory models have explained sound encoding at the cochlear level using static filter banks, but this vision is incompatible with the nonlinear and adaptive properties of the auditory system. Here we propose an approach that considers the cochlear processes as envelope interpolations inspired by cochlear physiology. It unifies linear and nonlinear adaptive behaviors into a single comprehensive framework that provides a data-driven understanding of auditory coding. It allows simulating a broad range of psychophysical phenomena from virtual pitches and combination tones to consonance and dissonance of harmonic sounds. It further predicts the properties of the cochlear filters such as frequency selectivity. Here we propose a possible link between the parameters of the model and the density of hair cells on the basilar membrane. Cascaded Envelope Interpolation may lead to improvements in sound processing for hearing aids by providing a non-linear, data-driven, way to preprocessing of acoustic signals consistent with peripheral processes.

[1] Aix Marseille Univ, CNRS, UMR7061 PRISM, UMR7020 LIS, Marseille, France. [2] Institute of Language, Communication, and the Brain (ILCB), Marseille, France. [3] CNRS, Aix Marseille Univ, UMR 7061 PRISM, Marseille, France. ✉email: etiennethoret@gmail.com

What and how do we hear? Sound waves are transformed into electrical signals through the interactions between the basilar membrane and the inner and outer hair cells, a fragile process that occurs at the first stages of the auditory system. Modeling these hearing processes through sound signal-processing models is still a timely question, in particular for curing hearing deafness through cochlear implant technologies. Cochlear implants indeed still fail to accurately recreate sounds with high fidelity due to limitations in replicating the mechanical-to-electrical transduction that occurs at the cochlear level.

Two opposite theories have influenced the development of signal-processing models of cochlear processes. The theories in question had their origin as models for pitch perception and have a well-established history within the discipline. On the one hand, Seebeck[1] put forth a temporal coding approach and showed that the pitch of a complex tone is dependent on repeated temporal signal patterns. On the other hand, Helmholtz[2] viewed the ear as a frequency analyzer implemented through the mechanical properties of the basilar membrane that processes the spectral components of sounds, known as the place coding theory. This theory, which was later supported by von Békésy's work on the physiology of hearing[3], has remained dominant and continues to influence the design of signal representations used in neuroprosthetic technologies such as cochlear implants.

According to this theory, the inner ear maps sound frequencies to specific places on the basilar membrane, leading to models based on linear spectro-temporal filter banks and Fourier analysis[4,5]. Despite its widespread agreement, this model-based view is still challenged by unanswered questions, conflicting views, and incompatible observations which still lead to vivid debates[6]. For instance, the sensitivity of the ear to sound signals has traditionally been measured using pure tones, resulting in equal-loudness curves[7] with a maximum sensitivity of around 4000 Hz. However, when the sound level of white noise, covering the whole frequency range, is reduced, neither its timbre nor its pitch changes[8], indicating the limitations of considering the ear as a -static- linear Fourier analyzer. These examples demonstrate the ongoing challenges and limitations in fully understanding and modeling cochlear processes.

Another phenomenon that is incompatible with the consideration of the cochlea as linear resonators are the perception of phantom sounds[9,10], such as the missing fundamental[11,12] and combination tones[13,14]. Although these phenomena have been thoroughly studied[2], their computational underpinnings remain unclear. Models based on linear Fourier analysis have incorporated posterior rectifications[15–18], but do not naturally account for these nonlinear phenomena. Whether combination tones and the missing fundamentals are caused by the same underlying phenomenon is still a source of debate[19], and no consensus has been reached on the origins of these nonlinearities in the auditory system. When no plausible interpretations are found at the peripheral processing level, such phenomena are often attributed to higher cortical processing and modeled using deep-neural networks[20]. There is currently no agreed-upon framework for explaining the generation of these nonlinearities in the auditory system.

What are we missing in the ear's sound processing? Model-based frequency decomposition, inspired by the dominant place coding theory, is not necessary to accurately simulate complex auditory tasks[21]. The auditory neural code extracted at the cochlear level is adaptively optimized to fit the acoustical structure of natural sounds, such as speech[22–24]. Temporal coding, based on timing cues also appears to play a crucial role after cochlear processes. Studies have demonstrated that amplitude modulations, driven by temporal coding, provide important acoustic cues both in the cochlea and auditory nerve[25–29]. These findings suggest that both temporal and place coding are present at the cochlear level in our hearing system, and are effective for different frequency domains. However, there is still a lack of a unified model that can reconcile these seemingly opposing behaviors. Some models have attempted to compensate for this discrepancy by adding nonlinearities to their spectral models[30], but these approaches do not provide an intrinsically unified explanation of the auditory processes.

In this paper, we tackle the question of modeling how the auditory system processes sound at the cochlear level. Rather than focusing on the resonances of the basilar membrane that inspired Fourier-based decompositions, we examine the sampling occurring at the level of the stereocilia, positioned at the top of the inner hair cell bundles. These stereocilia move in response to the endolymph motions, the fluid that fills the cochlea and which conveys vibrations, herewith performing interpolations of the incoming sound signal. To model the underlying cochlear processes, we propose to place envelope interpolation at the center of the sound coding. This computational model is inspired by the empirical model decomposition (EMD)[31,32] and interpolates a given signal based on the upper and lower signal envelopes. Upper and lower envelopes are envelopes of the signal obtained by interpolating respectively the maxima and minima of a signal. EMD offers a promising solution for methods that take into account hearing specificities in the case of noise[33], frequency selectivity[10,31], and source separation[34]. Our framework, therefore, provides an implementation that is compatible with a range of auditory adaptive and data-driven temporal coding phenomena. It can also account for nonlinear hearing behaviors. Furthermore, this model accounts for phenomena such as pure tone masking in noise[35], which is a canonical example of the cocktail party effect. It also provides a plausible explanation for adaptive coding[23] that occurs at the cochlear level. This framework is coherent with traditional cochlear filter properties and behaves as a constant-Q transform that takes into account the frequency selectivity of the ear. This also drives the perception of roughness, consonance, and dissonance of harmonic sounds[12].

This study embraces the challenge of modeling the human hearing sensory system. One issue of computational modeling is to accurately define the level of biological plausibility a model can afford and what it exactly accounts for. The relationship between biological observations and modeling is dynamic, with biological knowledge being influenced by the way we observe a phenomenon and modelers using functional interpretations of biological observations to inform their models. David Marr[36] proposed to categorize biological models into three levels—computational, algorithmic, and physical—to differentiate the relationship between the model from the underlying biological mechanism and from its computational function. Here, in the case of hearing, signal processing representations are used and can be placed at an intermediate level between the algorithmic and physical levels of Marr's framework. These signal representations provide a formal description of the encoding of sounds in the auditory pathway from the cochlea to the primary auditory cortex[37] and more recently up to cortical areas based on deep-neural network activations[38–43]. They are directly inspired by biophysical phenomena and measurement paradigms, such as auditory spectrograms that account for critical bandwidths of the basilar membrane or spectro-temporal modulation models based on neuronal responses in the auditory nerves and primary auditory cortex[37,44].

In the following sections, we detail the computational basis of this decomposition and demonstrate how it can be applied to modeling various psychoacoustic phenomena.

**Cascaded envelope interpolation**. Cascaded envelope interpolation (CEI) is a mathematical concept aiming to decompose signals into a finite set of modes, inspired by the EMD[32]. EMD consists of a cascaded process that extracts modes based on envelope interpolation and that shares striking similarities with auditory processes. Here, we introduce CEI, a variation of EMD with only one iteration and a fixed number of modes. This approach is different from EMD as the extracted mode does not fulfill the EMD's mode criteria.

The CEI starts by extracting upper and lower temporal envelopes from the original signal, averaging them to compute an interpolative envelope, and subtracting it from the original signal to get the first mode of the decomposition and a residual signal corresponding to the difference between the original signal and the mode. The residual is then used as an input for the next iteration to extract the next mode and so on until all the modes have been extracted (see Fig. 1a). This decomposition differs from the EMD algorithm in that each mode is extracted with only one interpolative envelope extraction and is not driven by a convergence threshold. In the framework of David Marr's theory, CEI is proposed as the algorithm that accounts for sound coding at the cochlear level.

It must also be noted that high-frequency modes are computed first and that higher-order modes correspond to lower frequencies. This behavior aligns with what is known about cochlear processes where the highest frequencies are extracted near the cochlear base while the lowest frequencies are extracted near the apex. The number of modes was arbitrarily fixed at 6 for each studied phenomenon as the energy of modes higher than 6 was close to 0.

CEI also differs from traditional envelope extraction, which often relies on the Hilbert transform of a signal, and which results in symmetric upper and lower envelopes. Conversely, CEI extracts upper and lower envelopes through a numerical process that involves finding the maxima and minima of the signal and averaging between them. This leads to an interpolative envelope that is not perfectly symmetric.

It is also important to note that the CEI is fully data-driven, as the maxima and minima depend solely on the signal. In order to compare this signal approach with traditional time-frequency representations, the power spectrum density of each mode can be computed and then summed up to provide a short-term Fourier transform (STFT) of the CEIs modes. It should be noted that this spectral analysis representation of CEI modes does not imply that a Fourier transform is performed in the auditory system. We only use it as a mathematical tool to compare the CEI with common signal representations. By considering that each mode is processed separately, its spectral content can be analyzed and used to compare with psychoacoustics results.

The CEI decomposition intrinsically differs from Fourier or Wavelet transforms as it is data-driven. Unlike traditional fixed dictionary transforms, CEI decomposes signals based on their own structure, without any prior assumption on the basis functions of the dictionary. For example, in the case of a signal composed of a chirp, a pure tone, and a modulated pure tone, CEI naturally separates each component (as shown in Fig. 1b), while classical models using fixed bandpass filter banks or Fourier-based decompositions fail to provide such a signal specific decomposition. In the traditional framework, source separation is achieved from complex models such as convolutional deep-neural networks fed by Fourier-based representations such as spectrograms.

Based on these observations and phenomenological considerations, we here challenge the ability of CEI to account jointly for different fundamental auditory phenomena: nonlinearities leading to phantom tone perception, frequency selectivity of the cochlear filter bank at the origin of roughness, consonance, and dissonance, and the data-driven behavior considering the sound processing at the cochlear level as an adaptive filter bank which can be revealed by frequency masking experiments.

## Results

**Phantom sounds and virtual pitches**. A vivid debate in hearing sciences lies in the origin of phantom sounds and virtual pitches which corresponds to situations where frequency components are perceived although they are not present in the Fourier spectrum[9]. These sounds, also known as combination tones (Fig. 2d), are created within the auditory system. They generally occur for signals composed of two pure tones with close frequencies, and have a lower frequency than the initial sounds[13,14]. Another close situation, known as the missing fundamental perception (Fig. 2a–c), occurs when the low frequencies of a harmonic signal are missing, but are still heard. This phenomenon can be observed when listening to the speaker of an old cellphone with a narrow frequency bandwidth. The fundamental frequency which is not physically present is virtually perceived and created within the auditory system allowing to perceive speech prosody. How many frequencies are created within the auditory system remains an open and still debated question. However, it is related to an essential non-linear mechanism that for instance plays an important role for communication in noisy environments to restitute masked frequency components.

Here we observe that CEI naturally produces these non-linear perceptual phenomena. The fundamental frequency of a speech sound is naturally reconstructed when low frequencies are removed by filtering (Fig. 2a, b). Similarly, CEI reproduces the most canonical combination tones which appear when two sinusoids of frequencies $f_1$ and $f_2$ are played at the same time ($f_1 < f_2$), and a third component at a frequency $2f_1$–$f_2$ is perceived (Fig. 2d, e and Methods). This phenomenon also appears in music where the combination of harmonic tones with distinct pitches are played at the same time, leading to a third perceived pitch (Fig. 2c). Combination tones were used by the baroque music composer Giuseppe Tartini[45] (Fig. 2f). These are striking demonstrations of the nonlinear behavior of hearing. While *a posteriori* rectifications are used to account for such behavior[15], we here observe that CEI naturally reveals these intra-aural generated components (Fig. 2a–e). Such a non-linearity is a direct consequence of the interpolation which becomes visible in the CEI spectrum. The fact that the CEI spectrum reveals components that are not present in the Fourier spectrum and that fit with those actually perceived suggests that this nonlinearity could occur directly at the hair-cell level as suggested by physiological observations[9]. Further physiological measurements are obviously necessary to confirm this claim and to make the CEI decomposition biologically plausible. This striking similarity suggests that hair-cell bundle motions, driven by the fluid motion triggering the spikes at the input of the auditory nerve, could be at the origin of such virtual sounds.

**CEI as an adaptive cochlear filter bank**. To satisfactorily serve as a candidate model for cochlear processes, CEI should also be compatible with the processing of the full set of sounds such as broadband noise, sound textures, and their mixing with speech and harmonic sounds such as music. Traditionally, the cochlea is believed to perform a constant-Q transform leading to representations such as auditory spectrograms[37]. In addition, cochlear processes are malleable and naturally adapt to the spectro-temporal content of incoming sounds[23]. We here tested the compatibility of CEI when considering the cochlea as an adaptive filter bank. It is indeed remarkable that the spectral shape of the

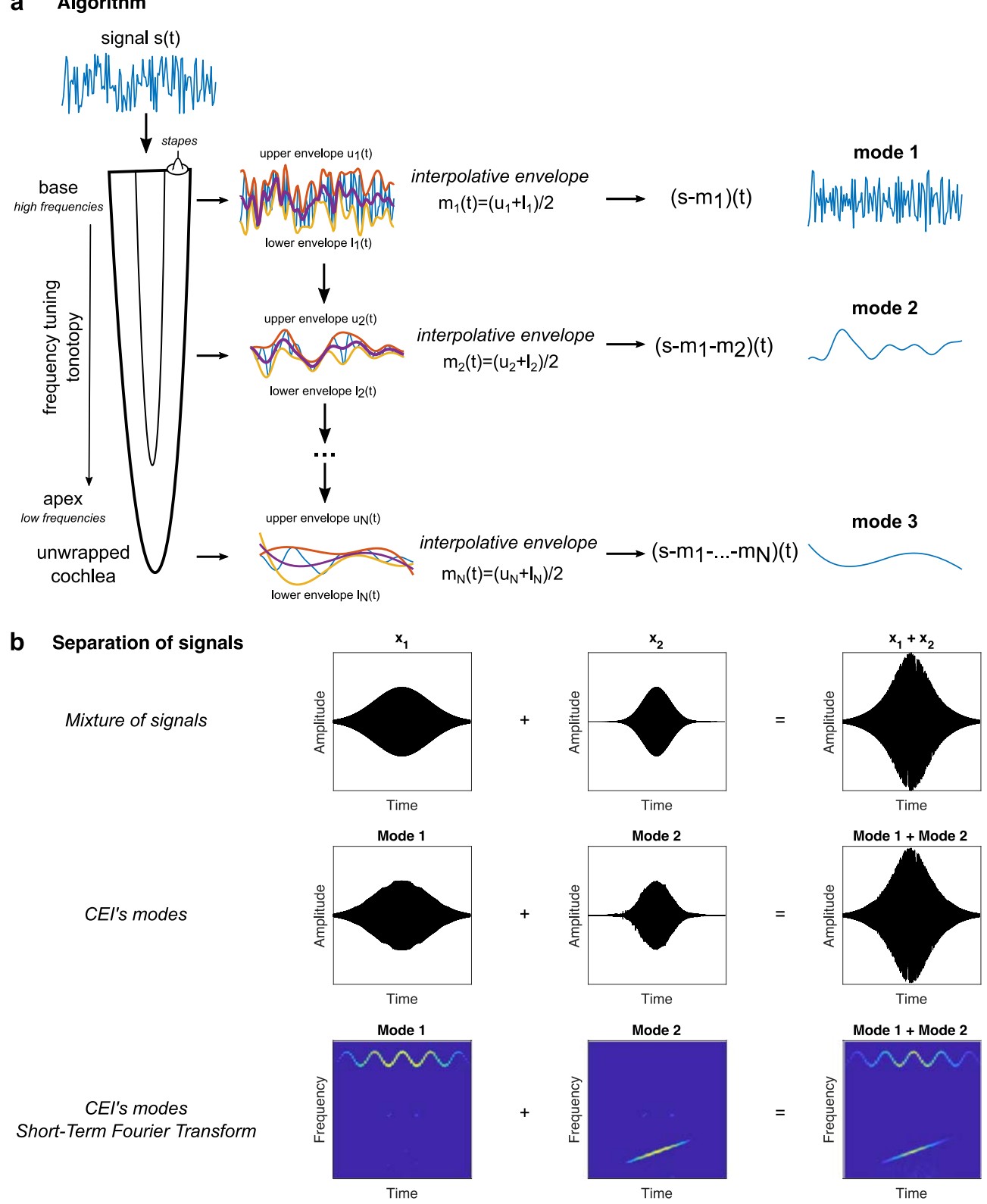

CEIs filter bank naturally adapts to the spectral content of the analyzed signal (Fig. 3a–c). We therefore here inquire whether CEI behaves as an adaptive filter bank whose properties fit with the equivalent rectangular bandwidth (ERB) model, the gold standard law accounting for the ears' bandpass processing.

Ears nonlinearly decompose sounds onto a code optimized for speech processing compatible with cochlear filters[46,47]. This behavior is often modeled as a band pass filter bank whose properties, i.e., central frequency and bandwidth are closely linked and follow a nearly linear relationship. This filter bank is adaptive which means that the filters automatically adapt their bandwidth according to the spectral content of a sound. Here we observe that CEI strikingly follows such an adaptive behavior with the same frequency bandwidth dependence as traditional

**Fig. 1 Cascaded envelope interpolation. a** The algorithm. The signal is analyzed through a finite iterative process. Local maxima and minima are first identified. Interpolative envelopes are further obtained by interpolators (here we used cubic spline) providing the upper envelope (in red) and the lower envelope (in yellow). The average between the two envelopes, the so-called interpolative envelope (in blue) is then obtained and subtracted from the original signal. The process is then repeated for a given number of modes. For each mode, the spectrum can be computed and then summed in order to compute the CEI's spectrum. This analysis can be done in the short term in order to compute the short-term CEI spectrum or spectrogram. These spectral representations are useful for comparison with Fourier representations. **b** Separation of tonal signal mixtures. A mixture signal (first row) composed of a frequency-modulated sinusoid ($x_1$) and a chirp ($x_2$) is analyzed with CEI. The two first modes are displayed (second row). To investigate the spectral contents of the modes, the short-term Fourier transform (third row) of each mode is then computed and summed to provide a representation of the combined spectral contents of the extracted modes. It is noticeable that CEI naturally extracts each signal component $x_1$ and $x_2$ separately in each mode.

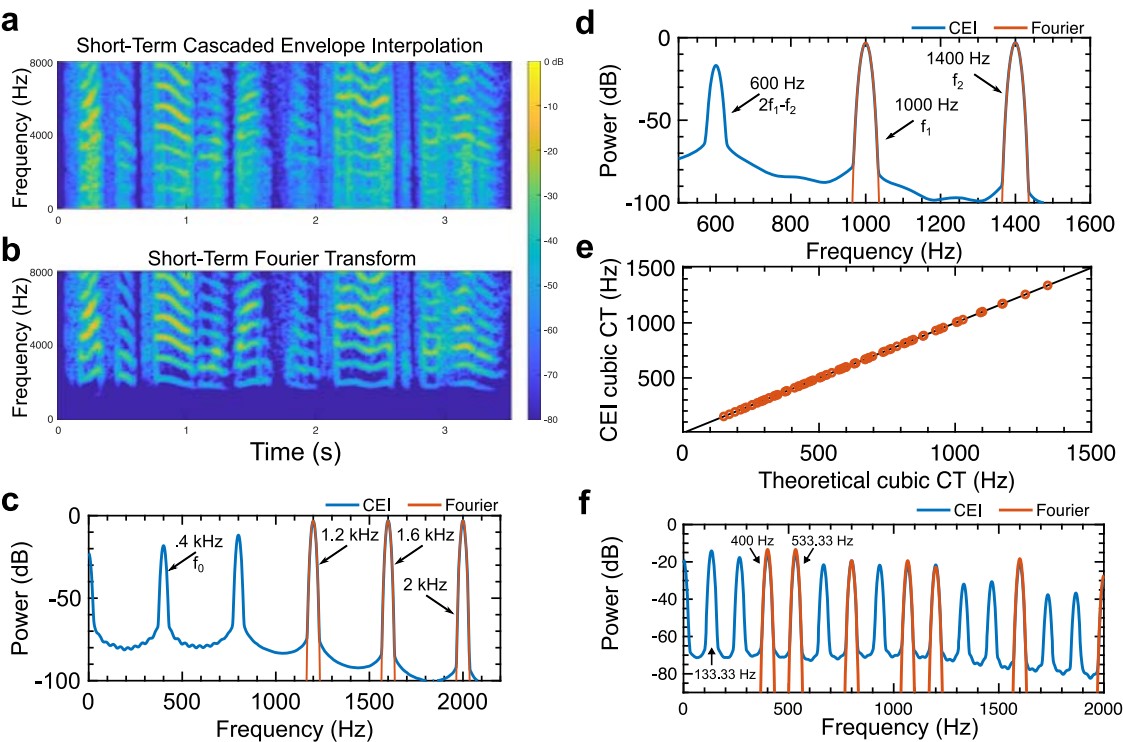

**Fig. 2 Cascaded envelope interpolation accounts for nonlinear phenomena.** Perception of the missing fundamental.Short-term Fourier spectrum of a speech signal without frequencies below 300 Hz (**a**) and short-term cascaded envelope interpolation spectrum (**b**). The figure reveals that CEI reconstructs the fundamental frequency just like the ear. **c** The Fourier spectrum (in red) and the spectrum of the cascaded envelope decomposition (in blue) for three synthetic pure tones. While the Fourier spectrum only shows the three components, the cascaded envelope decomposition reveals the missing spectral components which correspond to a signal with a fundamental frequency of 400 Hz which matches the perceived frequency. This illustrates how the CEI naturally accounts for virtual pitches while the Fourier spectrum does not. Combination tones. **d** One example of combination tones. Two pure tones of frequencies $f_1$ and $f_2$ lead to the perception of a third one of frequencies $2f_1-f_2$. While the Fourier spectrum does not reveal this phantom sound, the CEI spectrum has energy at this frequency. **e** Correspondence between the theoretical cubic combination tone frequency and the one predicted by CEI for 70 pairs of pure tones, see Methods for details. **f** Tartini sound. The Fourier spectrum (in red) and the spectrum of the cascaded envelope decomposition (in blue) of a combination of two synthetic harmonic sounds with fundamental frequencies 400 Hz and 533.33 Hz, the perfect fourth. While the Fourier spectrum provides the sum of the individual spectra, the cascaded envelope decomposition reveals components at 133.33 Hz and at 266.66 Hz which corresponds to the perceived phantom Tartini notes. The perceived pitch at 133.33 Hz corresponds to the difference between the two fundamental frequencies 533.33 Hz and 400 Hz.

cochlear filters[46,47]. We tested this by fitting a linear filter bank equivalent to CEI when processing a large database of speech sounds. This allowed us to compute the equivalent center frequencies and bandwidths for each CEI's mode considered as the result of a band-pass filter on the input signal (see Fig. 3d). Strikingly, the relationship between the center frequency and the bandwidth was compatible with psychoacoustics models based on the ERB. Such an adaptive behavior differs from current model-driven constant-Q transforms that consider the cochlea as a fixed band-pass filter bank. This makes it potentially compatible with adaptive efficient coding[22,23]. It is striking that CEI, like the auditory system, naturally adapts to the spectral content of a sound. CEI, therefore, opens a door to a computational basis for

the acoustical niches that may drive the co-adaptation between acoustic environments and the communications abilities of species. This indeed suggests that the auditory systems of living beings are based on an adaptive sensory coding, tuned to the communication signals of their conspecifics and to the environment, rather than a fixed model-driven filter bank.

**Frequency selectivity.** One other property of the cochlea is its ability to separate meaningful signals, often harmonic, in a mixture of noise with concurrent harmonic sounds. This sensory preprocessing is fundamental to perceiving speech in noise and is often known as frequency selectivity. This has historically been studied in canonical situations with pure tones masked by noisy

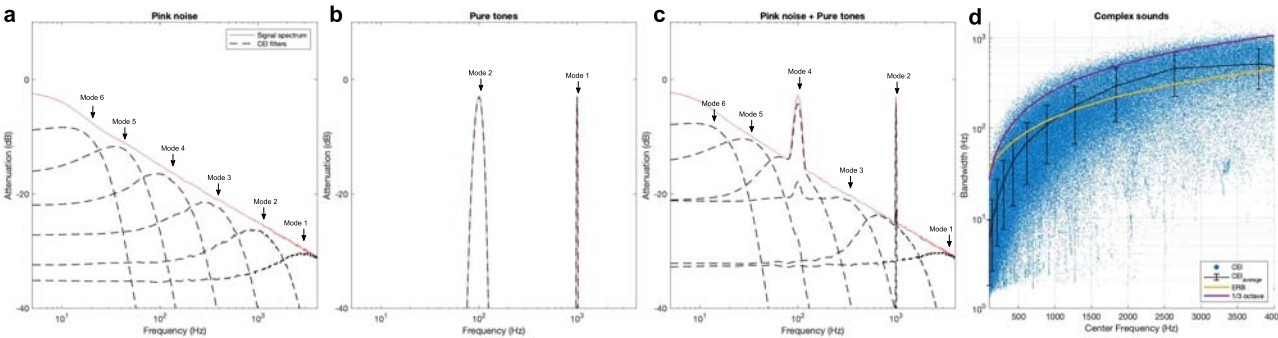

**Fig. 3 Cascaded envelope interpolation as an adaptive filter bank.** CEI naturally adapts to decompose the sound signal. **a** One hundred pink noises. **b** One hundred pure tones at 100 Hz and 1000 Hz. **c** A mixture between pink noise and pure tones has been analyzed with CEI. The spectrum of the 100 CEIs has been averaged. The filtering operated by CEI (dashed black lines) matches with the spectrum of the original signal (solid red line) and naturally adapts to the signal to be analyzed. **d** For complex sounds such as speech or environmental noise, the behavior of CEI can be interpreted in terms of equivalent band-pass filter bank properties (center frequency and bandwidth). CEIs equivalent linear bandpass filter bank (blue dots and black curve) follows the equivalent rectangular bandwidth ($r(410451) = 0.8$, $p < 10^{-10}$) and one-third octave $r(410451) = 0.8$, $p < 10^{-10}$) models corresponding to the theoretical bandwidth models of cochlear filters (yellow and violet curves). In order to give an idea of the bandwidth variability with respect to the center frequency, the black curve represents the mean and standard deviation on 11 points regularly spaced on the center frequency scale logarithmically sampled. Here we used a corpus of speech and environmental sounds[71]. For each sound, the CEI decomposition is first computed and the spectra of the equivalent linear filtering allowing to compute each mode from the original signal are then computed (see Method for details). The center frequencies and the bandwidths are finally computed and plotted (blue dots). As a comparison, the 1/3 octave model (in violet) and the ERB (in yellow) are superimposed and are coherent with the impulse response properties of the CEI equivalent linear bandpass filter bank.

mixtures[35] or by other pure tones[48]. Participants had to detect a probe signal in a given mixture in order to fit the bandpass filter properties. Here we reproduced these two situations in order to test whether CEI has the same frequency selectivity.

*Noise masker.* The human ability to detect sinusoids in noise is characterized by two main aspects: (1) The larger the noise frequency bandwidth the weaker the ability to detect the sinusoid. (2) The higher the frequency of the probed sinusoid the larger the bandwidth, also known as the auditory critical bandwidth. More precisely, above 500 Hz, the critical bandwidth increases linearly with respect to the logarithm of the sinusoid's frequency. We here wanted to test whether CEI reproduces these phenomena. In Patterson's masking experiment[35] sinusoids of a given frequency $f_0$ are played back simultaneously with a bandpass-filtered white noise centered at $f_0$ with a given bandwidth $\Delta f$. The noise level is increased until the subject is unable to detect the sinusoid, and the corresponding detection threshold is then determined accordingly. We simulated such an experiment for 5 frequencies $f_0$ (250 Hz; 500 Hz; 1000 Hz; 1500 Hz; 2000 Hz) and 15 frequency ratios $\Delta f/f_0$ between 0 and 1. We applied CEI for different bandwidths to determine a detection threshold of the sinusoids. Subjects were replaced by a virtual listener whose responses were simulated by CEI representations and a detection process. For a given frequency and a given ratio, the masking threshold was estimated thanks to an adaptive staircase procedure. CEI reproduces the expected behaviors (see Fig. 4). In particular, we fitted threshold curves similar to known psychophysical tuning curves obtained in perceptual experiments on humans[49]. This result confirms that CEI is compatible with the behavior of cochlea as a filter bank, interestingly the bandwidth at −3 dB of the corresponding filter is proportional to the central frequency.

*Pure tone masker.* CEI hence naturally accounts for the frequency masking of a pure tone in noise. This is particularly important for the perception of meaningful signals such as speech embedded in a mixture of background noise. However, frequency masking appears also in contexts such as music when pure tones are interacting. In the canonical context of the combination of two

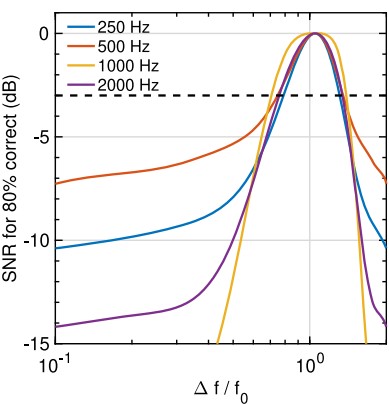

**Fig. 4 Masking thresholds of pure tones in noise.** Masking thresholds of sinusoids with respect to the normalized bandwidth $\Delta f/f_0$ of notched noise. We report signal-to-noise ratios (SNR) between the sinusoid and the notched noise that leads to 80% correct responses. For each frequency, the raw curves have been fitted with 10th-order polynomials. For each of the five frequencies, we observe that the normalized bandwidths (cutoff at −3 dB) of the corresponding bandpass filters are almost the same. This confirms that CEI is compatible with classical knowledge of auditory frequency masking.

pure tones, these beats lead to a sensation called roughness, which also relates to the notion of consonance or dissonance related to the frequency selectivity of the cochlear filter bank. In the speech, roughness drives the perception of aversiveness[50]. Formally, auditory roughness can be described as the perception of very fast fluctuations in sounds. It is now well-known that for stimuli composed of two pure tones, i.e., two tones that each have a single frequency, the sensation of roughness is driven by the ratio between the frequencies of the components. When the ratio is close to one, the perception leads to slow beats perceived as one single signal. When the ratio increases, the perception gives rise to a rough sensation revealing that the auditory system is unable to disentangle the two components. When the ratio further increases and reaches the critical bandwidth, the ears separate the components, and two frequencies are perceived[12]. From a

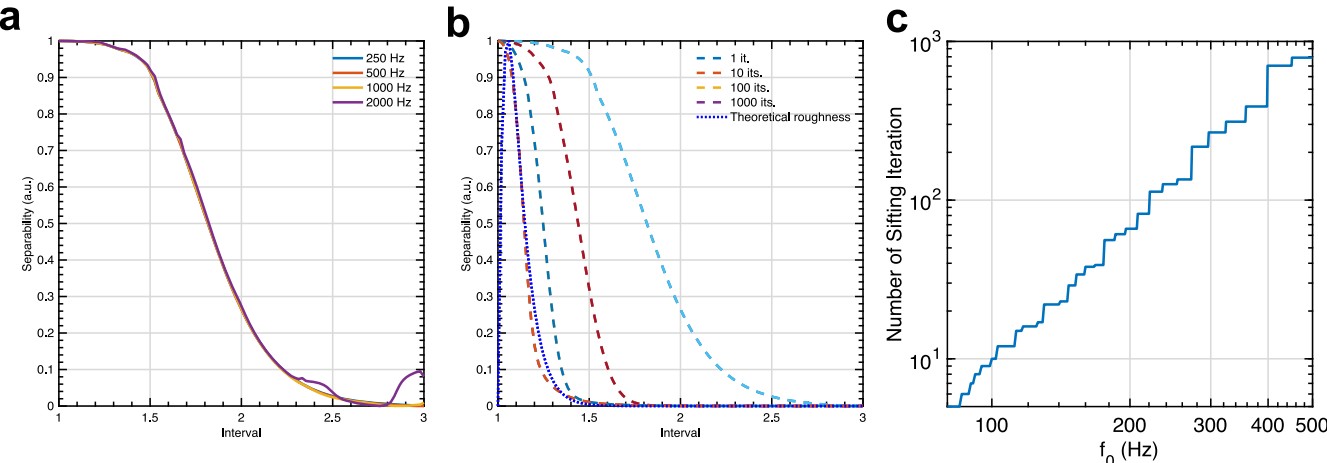

**Fig. 5 Separability of two pure tones. a** CEI separates a sum of two frequencies onto one slowly modulated frequency or two frequencies or an intermediate situation that can be assimilated to auditory roughness perception where it does not separate the two pure tones. The graphic shows the value of the separability index for 5 different frequencies $f_0$ (250 Hz, 500 Hz, 1000 Hz, 1500 Hz, 2000 Hz), and 50 intervals, i.e., ratio values $\alpha$. The frequency of the second component is computed as follows: $f_1 = \alpha f_0$, resulting in a signal: $s(t) = \cos(2\pi f_0 t) + \cos(2\pi f_1 t)$. **b** The CEI separability curves (dashed lines) for different numbers of sifting iterations and the theoretical roughness curves for $f_0 = 500$ Hz (dotted line). **c** The number of sifting iterations can be adjusted so that the separability index fits at best with the theoretical roughness curve. It is observed that the number of iterations necessary for the optimization increases as the frequency $f_0$ increases logarithmically. The number of iterations are necessary to fit the roughness curve is determined by minimizing the error between the frequency of the maximum value of the theoretical roughness curve and the frequency of the simulated separability curve where the separability is starting to differ from 1, here arbitrarily set to 0.98.

mathematical point of view, the sum of two pure tones can indeed be interpreted either as one sinusoid modulated by a slow modulation whose frequency is driven by the difference between the two components, or it can be seen as the sum of two components. Said differently, below a given frequency ratio, a sensation of roughness is perceived due to the ears' inability to disentangle the two pure tones. Here we aimed to test whether CEI behaves as hearing when decomposing a sum of two pure tones. We simulated situations in which CEI analyzes pairs of sinusoids and evaluated whether one or two frequencies were detected. When the two sinusoids are well separated by CEI, the first and the second mode of the decomposition respectively correspond to the first and the second sinusoid. But when the separability diminishes, either the third mode contains energy that can be related to the sensation of roughness, or the first mode corresponds to a sinusoid slowly modulated by another one. We here define an index of separability $d$ allowing us to characterize these three phenomena: when $d = 1$, the CEI considers the sum of sinusoids as one component, when $d = 0$, the CEI considers that the signal is composed of two separate sinusoids when $d$ is between 0 and 1, the CEI is not able to separate between the two pure tones and beats/roughness appear. Figure 5 shows the value of this separability index for different frequencies $f_0$, $f_1$, and ratios $\alpha$ with $f_1 = \alpha f_0$. We observe that the three behaviors are compatible with human auditory perception. As for the masking with noise, it is noticeable that the CEIs separability ability does not depend on the frequency while it is known that roughness maxima change with frequencies. This is coherent with the fact that CEI acts as a constant-Q filter bank in the auditory system. However, one current limitation of the model is that the frequency selectivity doesn't match with the theoretical roughness curves, see Fig. 5a.

An algorithmic way to address this limitation is to consider the EMD in its original form which includes a supplementary process, called sifting, which is an iterative process within each mode of computation (see Methods). However, there is currently no sound hypothesis on how a sifting process could be done by cochlear processes, or after. We removed this process by keeping only 1 iteration for each mode which makes the CEI process more

potentially implementable from a biological point of view by hair cells stereocilia. On the contrary, an iterative process with a threshold has no obvious plausible implementation. Nevertheless, by considering this process, we observe that the number of sifting iterations controls the frequency selectivity which can be adjusted to be more or less important according to the lowest frequency $f_0$. The higher the number of sifting iterations, the tighter the frequency selectivity and therefore the lower the roughness maxima (see Fig. 5b). We hereby determined the number of sifting iterations necessary to fit the index of separability with a roughness maximum similar to the theoretical curve for different frequencies $f_0$. We found a logarithmic relationship between the number of sifting iterations and the frequency $f_0$. Although its implementation in the cochlea remains very speculative, the number of iterations necessary to fit with the theoretical roughness is coherent with the inner hair cell distribution on the cochlea. This distribution is almost constant from the base to the apex, but as the frequency tuning varies logarithmically with respect to the cochlear tonotopy (Fig. 5c)[51], the number of inner hair cells involved per frequency band increases logarithmically with respect to the cochlear tonotopy. Such a correlation would suggest a possible link between the number of sifting iterations and the number of involved hair cells.

Interestingly, it is known that frequency selectivity decreases with aging because of deficient or damaged hair cells which leads to sensorineural hearing loss[52], especially for high frequencies. The origin of this larger bandwidth due to hearing loss remains debated, but the reduction of the number of healthy hair cells are known to impact such hearing damage. Here, we provide a concrete possible perspective to link the number of healthy hair cells with the increase of the bandwidth of cochlear filters (see Fig. 5b). This opens the possibility to use CEI as a model for sensorineural hearing loss. However, it remains necessary to more precisely understand how the sifting or an equivalent process could be implemented at a biological level.

Taken together, masking simulations with noise and pure tones reveal that CEI behaves as the auditory system for frequency selectivity. In addition, the effect of sifting iterations can reflect

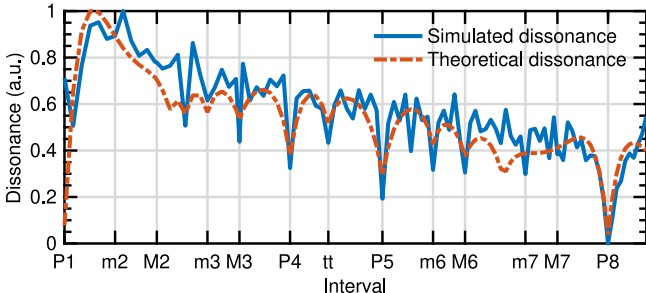

**Fig. 6 Dissonance curve of musical sounds.** The dissonance curve was obtained by simulating a pairwise comparison experiment based on the CEI spectrum of harmonic sounds (blue), and theoretical dissonance curves from Plomp and Levelt's model (red dashed), see Methods for details. A strong coherence between the theoretical consonance maxima and the simulated one is observed. This reveals that CEI accounts for the perception of consonance and suggests that sensory consonance can be processed at the very first steps of the auditory system.

the number of hair cells implicated in a mode extraction, as it is in accordance with cochlear tonotopy and with the effect of sensorineural hearing loss. This suggests a strong link between CEI and possible physiological implementation.

*Consonance and dissonance.* In relation to auditory roughness, when two musical instruments with two different timbre are playing the same note, roughness leads to beatings perceived as more or less consonant or dissonant[2,13]. Musical consonance is often associated with the pleasantness of a musical sound and conversely for dissonance. The origins of musical consonance and dissonance perception have been extensively studied from sensory and cognitive points of views[53–56]. In addition, models of consonance of complex sounds have been proposed based on the roughness curves obtained from pure tone combinations[12]. Here we tested the ability to predict theoretical consonance curves directly from the CEI representation. We simulated pairwise comparison experiments between pairs of tones with harmonic spectra (sawtooth). The underlying metrics used to simulate the pairwise judgments were based on the separability between the CEI spectra of the sum of the considered harmonic tones and the sum of the CEI spectra of each harmonic tone separately. We finally computed an arbitrary dissonance score characterizing the probability of a given interval to be judged as more dissonant than another one (Fig. 6). For the sake of coherence with the literature, dissonance curves have been transformed into consonance curves after being subtracted from 1. Interestingly, we observed that well-known consonant/dissonant intervals are naturally revealed by such an analysis. In particular, the octave (P8), the perfect fifth (P5), the perfect fourth (P4), the major sixth (M6), and the major third (M3) provide the most consonant intervals followed by the minor third (m3) and the tritone (tt). This result is well-known in music theory and confirms that CEI also aligns with this well-known auditory phenomenon. In this context, this would suggest that consonance/dissonance perception, which can also be driven by cultural and cognitive functions, would be mainly driven by bottom-up acoustic features as a consequence of envelope interpolation at the very first step of the auditory system.

## Discussion
We here present a data-driven framework based on a simple computational unit founded on CEI. By meta-analyzing well-known psychophysical phenomena in light of this transformation, we first show that it bridges linear, nonlinear, and adaptive

principles of peripheral hearing under a single framework. It supports that envelope extraction by interpolation is at the core of nonlinear and adaptive cochlear processes.

Envelope interpolation might be at the common origin of combination and phantom tones[19]. One current understanding of the missing fundamental suggests that autocorrelation is performed at the stage of the auditory nerve[51]. Such a process involves the implicit computation of time delays, which is still not elucidated and challenged by behavioral observations[57]. Combination tones have been considered as the consequence of different mechanisms such as nonlinear transduction at the hair-cell level[9] or due to central processes[58]. None of these models have to date provided a satisfying and unifying account of these psychophysical observations. In particular, whether peripheral and/or central processes are at the origin of these phenomena remains debated. On the other hand, CEI does not necessitate such a time delay computation hypothesis. If our study does not yet provide a precise account of how the complete mode extraction might be performed physiologically, the envelope interpolation is compatible with known intracellular recordings inside inner hair cells[59]. An important perspective is a better understanding of the active mechanism of outer hair cells and in particular distortion products leading to otoacoustic emissions[60]. The emitted frequencies corresponding to cubic distortion products could also potentially be explained by envelope interpolation distortions. Of course, this result doesn't preclude that the central system also contributes to these phenomena and the interplay between the peripheral and the central level of the auditory system remains to be clarified. Phenomena such as the missing fundamental perception could also be generated at higher levels of the auditory system[61]. Conversely, adding adaptivity and non-linearity in peripheral transformation modeling can also help us refine the understanding of the processing of different acoustical patterns such as spectro-temporal modulations[37] that are central for the perception of speech[62] and musical instrument timbre[63,64].

CEI offers a perspective on hearing adaptability by demonstrating that its computationally simple approach is compatible with the adaptive properties of hearing. The efficient coding kernel functions naturally comply with model-based cochlear filters[23]. However, the mechanism for extracting adaptive codes within the auditory system is not yet known. We suggest that envelope interpolation may be at the core of this process, as it has a higher physiological plausibility than the matching pursuit algorithms used to derive the efficient auditory code[23], which is not implementable at a physiological level. In addition, CEI not only accounts for phenomena ignored by linear auditory models but also simulates well-known phenomena such as the masking of pure tones in noise, auditory roughness, and musical consonance perception. This unification under a single data-driven framework opens avenues for reconsidering still-misunderstood phenomena, such as the cocktail party effect and how the brain tracks meaningful signals in noisy backgrounds.

CEI provides also a perspective on modeling sensorineural hearing loss resulting from hair cell damage or deficiency. We observed that the increasing critical bandwidth associated with sensorineural hearing loss can be accounted for by the number of sifting iterations, although the exact implementation of these iterations at the cochlear level is unknown. One possible implementation is a joint operation of a population of hair cells coding at the same location on the cochlea and computing the iterative envelope by averaging their activity. However, this remains a major limitation of the current model, and future research may help to make it even more biologically plausible.

Understanding the computational mechanism behind CEI could have implications for hearing aids. Hearing aids and cochlear implants still use passive gamma tone filter banks as

front-end representation, however, it is well established that temporal fine structures and temporal envelopes are essential for speech perception[22]. Here, we demonstrated that CEI can replicate the effects of hair-cell loss, which leads to an increase in the cochlear filter bandwidth. CEI thus offers a perspective on signal coding through implant electrodes, allowing for direct accounting of nonlinear cochlear behaviors while remaining compatible with cochlear filtering for non-stationary signals. In a broader context, efficient coding is also a fundamental principle of visual coding[65], and CEI may provide insights into how optimal codes can be extracted from other sensory systems such as vision.

From a larger theoretical perspective, CEI offers the possibility to reconsider the cortical processes of speech occurring after the cochlea in the primary and the first steps of the auditory system considering this unified data-driven framework. For speech perception, it has been shown that the brain tracks the sound envelope which is reminiscent of neural oscillations[66]. This has been interpreted as a functional principle to process speech signals by extracting the relevant sensory-motor oscillations, however, the computational bases of speech envelope modulations extraction from spectro-temporal information remains unclear. Considering CEI as the input, envelope extraction by interpolation can be achieved at very early processing stages and provides a plausible peripheral mechanism that supports and bridges with the existing literature on neural oscillations.

## Methods
All sounds were sampled at 16,000 Hz.

**Cascaded envelope interpolation**. CEI is a simplified version of the EMD[32] with only one envelope extraction at each step. It is interesting to note that this decomposition provides a perfect reconstruction of the original signal by simply summing up the modes. For a given signal, at each iteration, the upper and lower envelopes of the signal are extracted and averaged into an interpolative envelope which is then subtracted from the original signal. The interpolation is mathematically achieved by cubic spline interpolators. The process is then repeated a given number of times according to the number of defined modes, here 6 times. See Supplementary Fig. 1 for details.

**Short-term CEI**. In the spirit of time-frequency representations, we here define the short-term CEI as the time-frequency representation of the different modes of CEI. For each mode, the STFT is computed and then summed up together to form the short-term CEI which allows comparison with the STFT of the initial signals.

**Phantom sounds and virtual pitches**
*Combination tones.* Sums of two sinusoidal signals are generated with frequencies $f_1$ and $f_2$ defined such as $f_1 < f_2 = \alpha f_1$ for $7f_1$ values (500 Hz; 750 Hz; 1000 Hz; 1250 Hz; 1500 Hz; 1750 Hz; 2000 Hz) and $10\alpha$ values (1.3300; 1.3711; 1.4122; 1.4533; 1.4944; 1.5356; 1.5767; 1.6178; 1.6589; 1.7000). For each of the 70 pairs of sounds, we determined the value of the generated combination tone's frequency as the frequency with the biggest energy in the spectrum below $f_1$ or above $f_2$. This value was determined with an automated peak detection algorithm and was compared to the theoretical combination tone frequency with a regression.

**Adaptive filter bank**
*Pure tones + pink noise.* Tests signals composed either of pink noise, i.e., a random signal with a spectrum defined by $S(f) = 1/f^2$ and a random phase, or two pure tones with frequencies at 100 and 1000 Hz. The CEI decomposition of each signal is first computed and the CEI power spectrum of each mode is then computed. The operation is repeated 100 times with 100 different pink noise excerpts. The power spectra of each mode are then averaged.

*Sound database.* Speech and environmental sounds are 1500 excerpts from the Making Sense of Sounds (MSoS) challenge which are excerpts from the Freesound database[67], the ESC-50 dataset[68]. We chunked the 1500 excerpts in segments of 800 ms leading to 148,500 short sound excerpts.

*Equivalent linear filter bank.* Each excerpt $e(t)$ of the 148,500 segments was first decomposed with the CEI in 6 modes. Each mode $m(t)$ was then considered as the result of a linear convolution between the excerpt and a filter with a transfer function $h(t)$:

$$m(t) = (e * h)(t) \tag{1}$$

$h(t)$ was fitted by computing the cross-correlation between $m(t)$ and $e(t)$. In order to evaluate the correspondence of such a filter bank with the properties of the cochlear filter bank model, each transfer function $h(t)$ was modeled as a bandpass filter whose central frequency and bandwidth were determined based on its transfer function, see Supplementary Fig. 2.

**Frequency selectivity**
*Noise masker.* We simulated such an experiment for 5 frequencies $f_0$ (250 Hz; 500 Hz; 1000 Hz; 1500 Hz; 2000 Hz) and 15 frequency ratios $\Delta f/f_0$ between 0 and 1. We applied CEI for different bandwidths to determine a detection threshold of the sinusoids. Subjects were replaced by a virtual listener whose responses were simulated by CEI representations and a detection process described below. For a given frequency and a given ratio, the masking threshold was estimated thanks to an adaptive staircase procedure. Specifically, to simulate the detection task, we computed the CEI decomposition of each sinusoid + noise mixture and the minimum Euclidean distance between the target sinusoid and each CEI mode. We then only kept the minimum value of these distances which is supposed to simulate the best ability to detect the sinusoid in the mixture. To test whether the sinusoid is detected or not, we secondly computed the Euclidean distance between each CEI mode and the same white noise, but this time without the target signal. We consider that the sinusoid is detected when the distance with the signal is above the correlation with noise. The noise power was adjusted so that the detection rate is 80% with an adaptive staircase method (3 down/1 up with an average of the 30 last reversals). For each frequency and each notched noise, we repeated the adjustment 10 times. The average SNR was then computed for each frequency and fitted with a 10th-order polynomial function.

*Pure tone masker.* Two-tone separation experiments are made with CEI. Signal with two pure tones of frequency $f_0$ and $f_1$ are generated and analyzed with CEI. In order to determine whether CEI separates frequencies in this manner, we first define an index of separability:

$$d = \frac{\|m_1(t) - \cos(2\pi\alpha f_0 t)\|_{L2}}{\|\cos(2\pi\alpha f_0 t)\|_{L2}} \tag{2}$$

where $m_1(t)$ is the first extracted mode of the CEI. $m_1$ indeed corresponds to the highest frequency of the two pure tones as CEI extracts first the highest frequency which is supposed to be equal to $\cos(2\pi f_0 t)$ when the two pure tones are separated.

*Sifting.* To fit the roughness curves with the theoretical ones, we introduce the sifting iteration, a parameter from the original version of the EMD, which is involved in the extraction of each mode. In CEI, we stop the interpolative envelope extraction after one iteration. The interpolative envelope is computed and then subtracted to the original signal to provide the first mode and the process is repeated on the interpolative envelope to compute the second mode, and so on. For each mode, EMD continues the process until the residual, i.e., the difference between the signal and the interpolative envelope, becomes completely flat or monotonic. In EMD, the flatness is defined by a threshold with no prediction on how many iterations are necessary to converge. This process whose complexity is unknown, called sifting, makes EMD different from CEI. Here, we tested the influence of the number of iterations on the simulated roughness curves by fixing it arbitrarily. See supplementary Fig. 3 for a detailed scheme of the process. We adjusted the number of necessary sifting iterations to fit the simulated roughness maximum to the frequency interval leading to a theoretical roughness maximum according to the Plomp[13] model. In the simulation, the maximum roughness value is reached when the separability index falls below the arbitrarily chosen value $d = 0.98$.

*Consonance and dissonance.* We considered pairs of harmonic tones composed of two sawtooth signals of fundamental frequencies $f_0$ and $f_1 = \alpha f_0$ with $\alpha$ corresponding to 104 ratios between the two frequencies. Typical ratios are denoted on the interval axis (abscissa): P1 (1:1—perfect unison, $\alpha = 1$, $f_1 = f_0$), m2 (16:15—minor second), M2 (9:8—major second), m3 (6:5—minor third), M3 (5:4—major third), P4 (4:3—perfect fourth), tt (7:5—tritone), P5 (3:2—perfect fifth), m6 (8:5—minor sixth), M6 (5:3—major sixth), m7 (9:5—minor seventh), M7 (15:8—major seventh), P8 (2:1—perfect octave, $\alpha = 2$, $f_1 = 2f_0$). $13f_0$ were considered (261.63 Hz; 277.18 Hz; 293.66 Hz; 311.13 Hz; 329.63 Hz; 349.23 Hz; 370.00 Hz; 392.00 Hz; 415.31 Hz; 440.00 Hz; 466.17 Hz; 493.89 Hz; 523.26 Hz). Ten thousand pairwise comparisons were then simulated by computing the following separability index between the averaged CEI spectra of the 2 signals $s(f_0,t)$ and $s(\alpha f_0,t)$:

$$d = mse(\log 10(CEI(s(f_0, t) + s(\alpha f_0, t))), \log 10(CEI(s(f_0, t)) + CEI(s(\alpha f_0, t)))) + \varepsilon \tag{3}$$

where $mse$ is the mean square error and with $\varepsilon$ a Gaussian random noise added to the separability to arbitrarily introduce variability in the decision. A win-matrix corresponding to the times a given interval was found more dissonant than one

other was then computed, and a consonance/dissonance score was finally computed thanks to the Bradley–Terry algorithm[69].

**Reporting summary**. Further information on research design is available in the Nature Portfolio Reporting Summary linked to this article.

## Data availability

The sounds from the Making Sense of Sounds dataset can be accessed here: https://dcase-repo.github.io/dcase_datalist/datasets/sounds/msos.html or upon request. https://doi.org/10.17866/rd.salford.6901475.v4. The source data of all the figures are available here: https://doi.org/10.6084/m9.figshare.23264405. Please contact the corresponding author for any additional requests.

## Code availability

The custom codes to generate all the figures included in the paper can be accessed at the following repository: https://github.com/EtienneTho/hearing-as-cei/ and at this permanent link[70]: https://doi.org/10.5281/zenodo.8025054. The MATLAB scripts for computing the Cascaded Envelope Interpolation are provided here: https://github.com/EtienneTho/cei/ and at this permanent link[70]: https://doi.org/10.5281/zenodo.8025054.

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

## Acknowledgements
Research supported by grants ANR-16-CONV-0002 (ILCB), ANR-11-LABX-0036 (BLRI), and the Excellence Initiative of Aix-Marseille University (A*MIDEX).

## Author contributions
E.T., S.Y., and R.K.M. equally contributed to this work.

## Competing interests
The authors declare no competing interests.
