## [Peer Review File · Communications Biology]

This manuscript has been previously reviewed at another Nature Portfolio journal. This document only contains reviewer comments and rebuttal letters for versions considered at Communications Biology.

REVIEWERS' COMMENTS:

Reviewer #1 (Remarks to the Author):

The authors have done a very nice job of responding to my previous review. I think the explanation of the steps involved in the CEI analysis are now much clearer. I continue to think that the approach is interesting and that its success in explaining various phenomena is quite compelling. I thought the EEG results in the previous version were interesting – albeit not definitive. The authors have now removed that analysis. It is debatable as to whether that weakens or strengthens the manuscript. I think that either is fine and it depends on the authors. Overall I think the manuscript is interesting and almost ready for publication.

I have only a few very minor follow up comments/queries:

- 1) In figure 1 – in the top right panel it looks like Mode 1 is simply the original signal. But it should be the first interpolated envelope, no?
- 2) Assuming I understand the analysis correctly, it seems to me that the phrasing of the CEI process on lines 319-321 is badly worded. The average of the upper and lower envelopes is the first mode, right? And then you subtract that from the signal to get the residual. This is not how it reads on these lines.
- 3) Typo: lines 312-314 “Cascaded envelope interpolation [is] a mathematical concept aiming to decompose signals into a finite set of modes inspired by [] Empirical Mode Decomposition...”
- 4) The lines 366-368 say “Conversely, CEI extracts upper and lower envelopes through 366 a numerical process that involves finding the maxima and minima of the signal and computing separate interpolations between them.” Separate interpolations? This seems badly phrased to me too.
- 5) “To candidate as...” is unusual English. I suggest “To satisfactorily serve as a candidate model for cochlear processes...”
- 6) Line 533 “enquire” instead of “enquiry”
- 7) Line 575-576 “band-[p]ass”!

Reviewer #2 (Remarks to the Author):

I had only very comments on the first version of this paper, and all of them have been satisfactorily taken into account. I have no further remarks.

Reviewer #3 (Remarks to the Author):

Overall, this is a good article. It is well-written and presents a novel and interesting framework about

sound encoding in the early auditory system in a clear manner. The proposed model provides a satisfying and unifying account of many psychophysical observations. The authors consider alternative models fairly and propose a parsimonious and reasonable algorithm for modelling peripheral hearing that bridges linear, nonlinear, and adaptive principles. The authors adequately acknowledge the need for further physiological measurements to confirm the biological plausibility of the proposed algorithm, particularly regarding the nonlinearity that could occur directly at the hair-cell level. Nonetheless, the proposed algorithm exhibits very interesting processing properties and explains a broad range of psychophysical effects.

Despite not having access to the initial version of the manuscript, I have been thoroughly impressed with the quality of comments from the three initial reviewers, as well as the authors' responses and revisions. In particular, as I was tasked with evaluating the authors' rebuttal to Rev 3 from the initial submission, I paid careful attention to this aspect. After thorough evaluation, I am confident that the revisions effectively tackle the issues highlighted by Rev 3, and I am pleased to report that the updated article is sufficiently straightforward to comprehend, with the primary arguments being easy to follow and understand. The figures serve as an invaluable aid in understanding the fundamental concepts behind the new model proposed by the authors.

With regard to Rev 3's observations on the role of downstream auditory system levels in the encoding process, I concur that the authors could have better acknowledged their possible implication. Although the authors rightly noted in the discussion that "this result doesn't preclude that the central system also contributes to these phenomena," I believe that further elaboration on this aspect would be beneficial. Specifically, given the iterative nature of the encoding process, it would be worthwhile for the authors to mention, even briefly, how the proposed mechanism may or may not implicate, or be influenced by, higher levels of the auditory hierarchy.

Of note, regarding speech emotions (P14, L645), the article states that roughness drives the perception of speech emotions based on previous research. However, given that speech emotions can be driven by various aspects of speech (such as prosody and semantics), it may be helpful to rephrase the sentence to more directly refer to the aversive or alarming nature of this acoustic feature.

There are also a few typos: "We therefore here enquiry" on page 17 line 251 should be revised, and the first word in line 445 'representation' is out of place.

⇒ We would like to thank again the reviewers for their helpful and relevant remarks that helped us to improve the manuscript a great deal along with the different rounds of review.

Reviewer #1 (Remarks to the Author):

The authors have done a very nice job of responding to my previous review. I think the explanation of the steps involved in the CEI analysis are now much clearer. I continue to think that the approach is interesting and that its success in explaining various phenomena is quite compelling. I thought the EEG results in the previous version were interesting – albeit not definitive. The authors have now removed that analysis. It is debatable as to whether that weakens or strengthens the manuscript. I think that either is fine and it depends on the authors. Overall I think the manuscript is interesting and almost ready for publication.

I have only a few very minor follow up comments/queries:

1) In figure 1 – in the top right panel it looks like Mode 1 is simply the original signal. But it should be the first interpolated envelope, no?

⇒ We have double checked and although they look close, mode 1 and the original signal are different.

2) Assuming I understand the analysis correctly, it seems to me that the phrasing of the CEI process on lines 319-321 is badly worded. The average of the upper and lower envelopes is the first mode, right? And then you subtract that from the signal to get the residual. This is not how it reads on these lines.

⇒ The process is actually this one and was misleading in the first version of the manuscript which may have induced a misunderstanding: the N-th mode corresponds to the original signal minus the sum of N-th previous interpolative envelope. It can be double checked in the Supplementary Figure 3 which describes the original EMD process, if you consider an EMD process with one iteration, as we define CEI, the residual is set to the signal at the first iteration and the interpolative envelope is then computed with only one iteration and then subtracted to the residual to provide the first IMF. For CEI and for the first mode it corresponds to subtract the first interpolative envelope to the initial signal.

⇒ Along with this response, we noticed a remaining small mistake in the Supplementary Figure 1 which we've now corrected.

3) Typo: lines 312-314 “Cascaded envelope interpolation [is] a mathematical concept aiming to decompose signals into a finite set of modes inspired by [] Empirical Mode Decomposition...”

⇒ Done

4) The lines 366-368 say “Conversely, CEI extracts upper and lower envelopes through 366 a numerical process that involves finding the maxima and minima of the signal and computing separate interpolations between them.” Separate interpolations? This seems badly phrased to me too.

⇒ We have rephrased as follows: “[...] CEI also differs from traditional envelope extraction, which often relies on the Hilbert transform of a signal, and which results in symmetric upper and lower envelopes. Conversely, CEI extracts upper and lower envelopes through a numerical process that involves finding the maxima and minima of the signal and averaging between them. This leads to an interpolative envelope that is not perfectly symmetric. [...]”

5) “To candidate as...” is unusual English. I suggest “To satisfactorily serve as a candidate model for cochlear processes...”

⇒ We thank the reviewer for this suggestion.

6) Line 533 “enquire” instead of “enquiry”

⇒ done

7) Line 575-576 “band-[p]ass”!

⇒ done

Reviewer #2 (Remarks to the Author):

I had only very comments on the first version of this paper, and all of them have been satisfactorily taken into account. I have no further remarks.

Reviewer #3 (Remarks to the Author):

Overall, this is a good article. It is well-written and presents a novel and interesting framework about sound encoding in the early auditory system in a clear manner. The proposed model provides a satisfying and unifying account of many psychophysical observations. The authors consider alternative models fairly and propose a parsimonious and reasonable algorithm for modelling peripheral hearing that bridges linear, nonlinear, and adaptive principles. The authors adequately

acknowledge the need for further physiological measurements to confirm the biological plausibility of the proposed algorithm, particularly regarding the nonlinearity that could occur directly at the hair-cell level. Nonetheless, the proposed algorithm exhibits very interesting processing properties and explains a broad range of psychophysical effects.

Despite not having access to the initial version of the manuscript, I have been thoroughly impressed with the quality of comments from the three initial reviewers, as well as the authors' responses and revisions. In particular, as I was tasked with evaluating the authors' rebuttal to Rev 3 from the initial submission, I paid careful attention to this aspect. After thorough evaluation, I am confident that the revisions effectively tackle the issues highlighted by Rev 3, and I am pleased to report that the updated article is sufficiently straightforward to comprehend, with the primary arguments being easy to follow and understand. The figures serve as an invaluable aid in understanding the fundamental concepts behind the new model proposed by the authors.

With regard to Rev 3's observations on the role of downstream auditory system levels in the encoding process, I concur that the authors could have better acknowledged their possible implication. Although the authors rightly noted in the discussion that "this result doesn't preclude that the central system also contributes to these phenomena," I believe that further elaboration on this aspect would be beneficial. Specifically, given the iterative nature of the encoding process, it would be worthwhile for the authors to mention, even briefly, how the proposed mechanism may or may not implicate, or be influenced by, higher levels of the auditory hierarchy.

⇒ In order to answer this comment, we have now added the following paragraph: “[...] Of course, this result doesn't preclude that the central system also contributes to these phenomena and the interplay between the peripheral and the central level of the auditory system is still unclear. Phenomena such as the missing fundamental could also be generated at higher level of the auditory system (Chialvo, 2003). And conversely, these new lights on peripheral transformations can also help us to refine the understanding of the processing of different acoustical patterns such as spectro-temporal modulations (Chi et al., 2005) that are central for the perception of speech (Elliott & Theunissen, 2012) and musical instrument timbre (Patil et al., 2012; Thoret et al., 2021). [...]”

Of note, regarding speech emotions (P14, L645), the article states that roughness drives the perception of speech emotions based on previous research. However, given that speech emotions can be driven by various aspects of speech (such as prosody and semantics), it may be helpful to rephrase the sentence to more directly refer to the aversive or alarming nature of this acoustic feature.

⇒ We have now rephrased as follow: “[...] In speech, roughness drives the perception of the aversiveness [...]”

There are also a few typos: "We therefore here enquiry" on page 17 line 251 should be revised, and the first word in line 445 'representation' is out of place.

⇒ Done